# In Situ Sonification of Anaerobic Digestion: Extended Evaluation of Performance in a Temperate Climate

**John Loughrin \*** 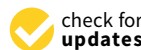**, Stacy Antle, Jason Simmons, Karamat Sistani and Nanh Lovanh**

United States Department of Agriculture, Agricultural Research Service, Food Animal Environmental Systems Research Unit, 2413 Nashville Road, Suite B5, Bowling Green, KY 42101, USA; stacy.antle@usda.gov (S.A.); jason.simmons@usda.gov (J.S.); karamat.sistani@usda.gov (K.S.); nanh.lovanh@usda.gov (N.L.)

\* Correspondence: john.loughrin@usda.gov; Tel.: +1-270-781-2260

**Abstract:** Increasing the efficiency of anaerobic digesters and improving sludge breakdown is vital to reducing the cost of biogas production and reducing the environmental consequences of sludge disposal. The performance of two unheated anaerobic digestion systems, one exposed to sound at <20 kHz by waterproofed speakers and one acting as a control, were compared for over a year. The digester systems were both composed of primary (11.4 m$^3$) and secondary (3.8 m$^3$) anaerobic tanks, facultative tertiary (3.0 m$^3$) tanks and an aerobic holding tank from which effluent was mixed with feed and recirculated back to the system. Exposure of the gas saturated digestate to a low frequency sine wave induced numerous bubble harmonics up to, and presumably beyond, ultrasonic range, showing that sonification of a highly gaseous liquid might be used to accomplish low power ultrasonication of digestate at greater distances than is possible with conventional ultrasonic technology. Through the summer of 2019, the sound-treated system produced 27% more biogas than the control system, and 74 times more during the winter when biogas production by the control systems essentially ceased. Afterwards, the control system produced more biogas due to depletion of volatile solids in the sound-treated digester. Results show that sound can be used for faster digester startup and substitute for a share of heating requirements during cool months.

**Keywords:** anaerobic digester; biogas; bubbles; carbon dioxide; cavitation; dissolved gas; greenhouse gas; methane; sludge; sonification; ultrasonication

## 1. Introduction

Waste generated by concentrated animal feeding operations (CAFO) is a challenging environmental problem. The large volumes of waste generated makes treatment prohibitively expensive and thus impractical by conventional means. This has led to extensive research into alternative means of treatment and disposal in the last few decades. Unfortunately, none of these have achieved widespread acceptance. Nevertheless, high daily production of organic matter coupled with excessive and unbalanced plant nutrient concentrations (N, P, K) as well as the possible presence of pathogens and antibiotic resistance genes continues to make finding solutions to treating animal waste imperative.

One of the most challenging aspects of wastewater treatment is the large volumes of organic sludge generated, whether directly by animals in CAFO, or produced by aerobic treatment of municipal wastewater [1]. In CAFO, sludge handling most often requires large volumes of water for flushing from feeding lots and housing, whereas in sewage treatment plants, excess generated sludge, composed of aerobic bacteria with complex cell walls and gelatinous extracellular matrices, often requires landfilling due to high concentrations of heavy metals [2]. In the former case, the flushed waste is typically stored/treated in facultative lagoons and subsequently applied to fields as fertilizer. This often leads to groundwater and surface water pollution due to leaching of organic matter and plant nutrients [3,4].



Sludge in wastewater can only be reduced by either removal (e.g., chemical flocculation followed by solids-liquid separation) [5] or hydrolysis to convert polymeric organic matter to soluble chemical oxygen demand (sCOD) which can then be reduced by aerobic or anaerobic treatment. In this latter case, hydrolysis is usually understood to be the rate-limiting step in sewage treatment, often leading to intentional uncoupling of hydraulic retention times (HRT) from sludge retention time (SRT) so that sludge may be retained, and hydrolysis performed more effectively [6].

In the case of both municipal and CAFO wastewater, therefore, ways of accelerating sludge hydrolysis are desirable. A number of means have been tried, including biological pretreatments such as adding hydrolytic enzymes or microorganisms capable of producing such enzymes. As noted by Yu et al. [7], however, the addition of exogenous enzymes is usually too expensive to be practical considering the high volumes of waste produced. For both adding enzymes and hydrolytic microorganisms, environmental conditions need to be considered, e.g., pH, temperature, and the presence of inhibitory substances since these conditions could affect the efficacy of sludge treatment [8,9].

Another means of reducing sludge and enhancing biogas production has been chemical hydrolysis with either acid or alkali [10,11]. Chemical hydrolysis has been shown to increase soluble chemical oxygen demand (sCOD) and methane production. Addition of these reagents to large volumes of sludge is expensive, however. One could speculate, nonetheless, that alkaline hydrolysis could be more economically favorable than acid hydrolysis due to the high wastewater bicarbonate ($HCO_3^-$) buffering rendering acidification difficult [12].

Pretreatment of sludge by ultrasonification has also received considerable attention [13]. By achieving partial disintegration of sludge, i.e., by reducing particle size and disrupting cell walls and extracellular biofilms, the biodegradability of wastewater can be enhanced and significant gains in biogas production achieved. Much of the effectiveness of ultrasonification for sludge reduction can be ascribed to acoustic cavitation, in which a bubble oscillates at resonant frequencies that are proportional to its radius. Although also termed "stable" cavitation due to an indeterminate period of quasi-stable, small amplitude oscillations about the resonant radius, the bubble will tend to grow and eventually collapse as the bubble pressure falls below a pressure lower than that of the external medium. The pressure amplitude caused by this cavitational collapse may be quite intense [14].

As an alternative to ultrasonic pretreatment of wastewater prior to anaerobic digestion, sound at sonic frequencies has been used as a continuous means of treatment in anaerobic digesters. For the purpose of this discussion, sonification will be used to refer to excitation at less than 20 kHz, while ultrasonification will be used to refer to excitation frequencies above 20 kHz. As sound intensity is generally attenuated with distance in proportion to the square of its frequency, the large size of digesters dictated the use of relatively low frequency sound. In a pilot-scale experiment using semi-continuous sound treatment, biogas production was more than doubled as compared to a control digester [15].

Figure 1 illustrates some of the phenomena that could be induced by acoustic excitation in gas-saturated anaerobic digestate. Acoustic streaming is a steady flow in which fluid and particles suspended in the fluid oscillate sinusoidally in the direction of the wave propagation. The components of the acoustic wave consist of a bulk flow velocity and a regular oscillatory motion; both of these can increase shear stresses but the latter component in particular increases cell membrane porosity and is used to enhance delivery of molecules to cells in the process termed sonoporation [16]. While ultrasonic frequencies are employed during sonoporation, bubbles undergoing linear oscillation in an acoustical field emit harmonics at twice to many times that of the excitation frequency [14]. These harmonics could potentially range into ultrasonic frequencies, meaning that sonification of a highly gaseous liquid can lead to low intensity ultrasonification at much greater distances from the emitter than is possible with primary ultrasonification.

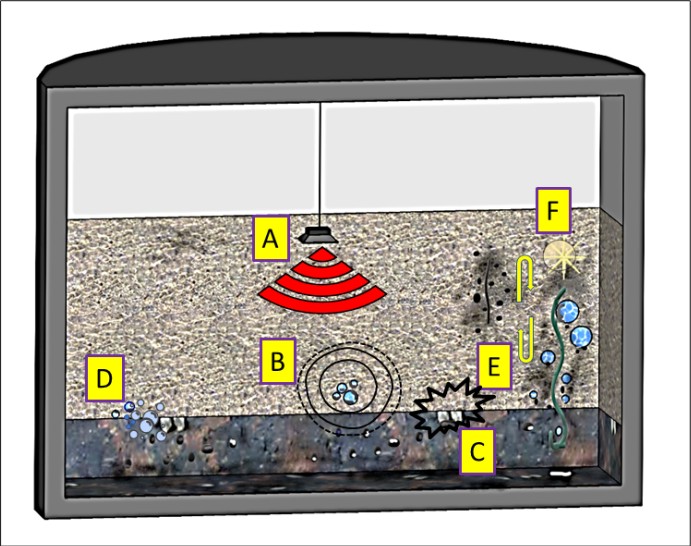

**Figure 1.** Representation of acoustically influenced processes in anaerobic digester. A. Acoustic streaming; B. Bubble harmonics; C. Vibrational energy imparted to sludge and suspended particles; D. Cavitational inception; E. Bubble drag and lift of solid matter; F. Cavitational collapse.

All of these factors could lead to improved sludge breakdown. Furthermore, gas production could be increased by accelerating the rate of ebullition as bubbles form in regions of reduced pressure and collapse at the surface of the digestate, in addition to the direct release of subsurface gas due to added kinetic energy imparted to digestate.

The oscillatory component of acoustic streaming induces vibrations in suspended particles. This can serve to reduce boundary layer thickness and enhance molecular exchange between microorganisms and particles and the bulk medium [17]. Some of the effectiveness of (ultra)sonification in sludge disintegration, though, can be attributed to cavitation. In an acoustic field, a fluid is subject to compressions and ratifications. In a liquid containing a mixture of gases, the gas can exist in a solvated state, i.e., by hydrogen bonding to water, or as non-hydrogen bonded free gas. Due to Brownian motion, sub-micron sized gas voids form in the water. Due to the ongoing cycle of pressure rarifications and compressions caused by an acoustic field, these voids grow in size and become nascent bubbles, which continue to oscillate and grow until buoyancy causes them to rise to the surface or to collapse when their internal pressure drops sufficiently below that of the external pressure [14].

The drag and lift forces generated by rising bubbles can also enhance mixing by increasing circulation within anaerobic digesters, a phenomenon that has been exploited to enhance biogas production by recirculating biogas through digesters [18].

Whether accomplished by sonication or ultrasonification, sludge reduction by sound excitation is likely due to mixed mechanisms. Forces due to cavitational collapse are more likely to predominate during ultrasonification pretreatment as higher pressures are generated by the collapse of small bubbles than by the collapse of bubbles with larger radii as shown in the Young–LaPlace equation [19]:

$$P_b = P_o + \frac{2\gamma}{R_B} \tag{1}$$

where $P_b$ equals the pressure within the bubble, $P_o$ equals the pressure of the external medium, $\gamma$ equals the bubble surface tension and $R_B$ is the bubble radius. Thus, the collapse of bubbles undergoing acoustic cavitation at ultrasonic frequencies will generate much more force than that of bubbles resonant at sonic frequencies. Cavitation is also more likely to occur during high-intensity ultrasonification than during lower-power sonication [20] although doubtlessly the gas saturated conditions of the wastewater in anaerobic digesters makes a high occurrence of cavitational events likely regardless.

In the case of waste sonification, while the forces generated by cavitational collapse of large bubbles are likely to contribute to sludge breakdown, the ongoing nature of the treatment makes the contribution of factors such as acoustic streaming, degassing of digestate, and circulatory forces due to bubble lift and drag also likely to play a significant role in waste breakdown. In the case of either ultrasonic pretreatment or sonic treatment, the goal is the same; reduction of sludge particle size facilitating microbial colonization and their accelerated growth.

Here, we report on the operation of unheated control and sound-treated anaerobic digestion systems operated yearlong in South-central Kentucky, which is characterized by a moist, subtropical climate with an average annual temperature of 13.8 °C, and an average of 1275 mm precipitation. Waste loading rates and feedstock were varied depending on weather conditions, the amount of biogas produced, and wastewater quality in order to prevent overloading of the systems. The systems could, thereby, be compared under varying environmental conditions and waste loading rates. Thus, it could be determined if sonic treatment of anaerobic digestate is a viable long-term approach for enhancing biogas production.

## 2. Materials and Methods

### 2.1. Digester and Audio Systems

The digester and audio systems have been described previously [15] except that Skar Audio FSX10-4 ten in (25.4 cm) 4 $\Omega$ speakers rated at 200 W RMS (root mean square) power (Skar Audio, Tampa, FL, USA) were used rather than eight in speakers. Briefly, the digesters consisted of two parallel rows of tanks consisting of 11.4 $m^3$ primary digesters, 3.8 $m^3$ secondary digesters, 3.0 $m^3$ holding/settling tanks, and 1.2 $m^3$ aeration tanks. The volumes of digestate in these tanks were maintained at approximately 9.1, 3.0, 2.3, and 0.8 $m^3$, respectively, by means of float switches and electrically-actuated valves. Water from the aeration tank was recycled to a 0.9 $m^3$ cone bottom feed mixing tank. Feed was added to 400 L of the recycled water, fed to the primary digesters and then rinsed with an addition 400 L of water. The order of feeding to the sound-treated or control digester systems was alternated with each feeding.

During a first experiment, the digesters had been operated from 1 May through 18 Oct 2018. From then until 7 May 2019, the digesters were not fed, and were opened to perform maintenance. The loudspeakers in replaced, and during May, each digester system was fed 90.7 kg cracked corn. The digester systems were not adequately sealed until June, however, and the evaluation period encompassed the months June 2019 through July 2020.

From then on, the digesters were fed on a varying schedule but more frequently in warmer months. The feed material varied and consisted of cracked corn, waste-activated sludge (WAS) from the Bowling Green Municipal wastewater treatment system, corn stover, poultry litter, shredded paper, and cardboard. The respective feedstocks had moisture contents of 14.4%, 68.3%, 28.6%, 24.6%, 4.1, and 10.0% moisture, and volatile solids (VS) contents of 97.1%, 65.7%, 83.0%, 77.2%, 83.0%, and 97.2%. Figure 2 shows the quantity and source of volatile solids fed each month, and selected weather conditions for the evaluation period.

Audio recordings were amplified once by applying a gain of 400% and converted to monaural MP3 files at a constant bit rate of 320 kbps. Noise reduction was performed in WavePad Masters Edition version 10.67 (NCH Software Inc., Greenwood Village, CO USA (www.nch.com.au)) by applying audio spectral subtraction using a silence to audio proportion of 20%. Data were plotted in Audacity version 2.2.2 (audacityteam.org) as fast Fourier transforms (FFT) using 4096 samples which, using a sampling rate of 44.1 kHz, gave a frequency resolution of 10.8 Hz, and plotted on a linear scale in a Hanning window.

Audio amplifiers were operated at one-eighth volume until 9 July 2019 when investigation into the effect of sound intensity was begun and at one-half volume from 8 October 2019 onwards.

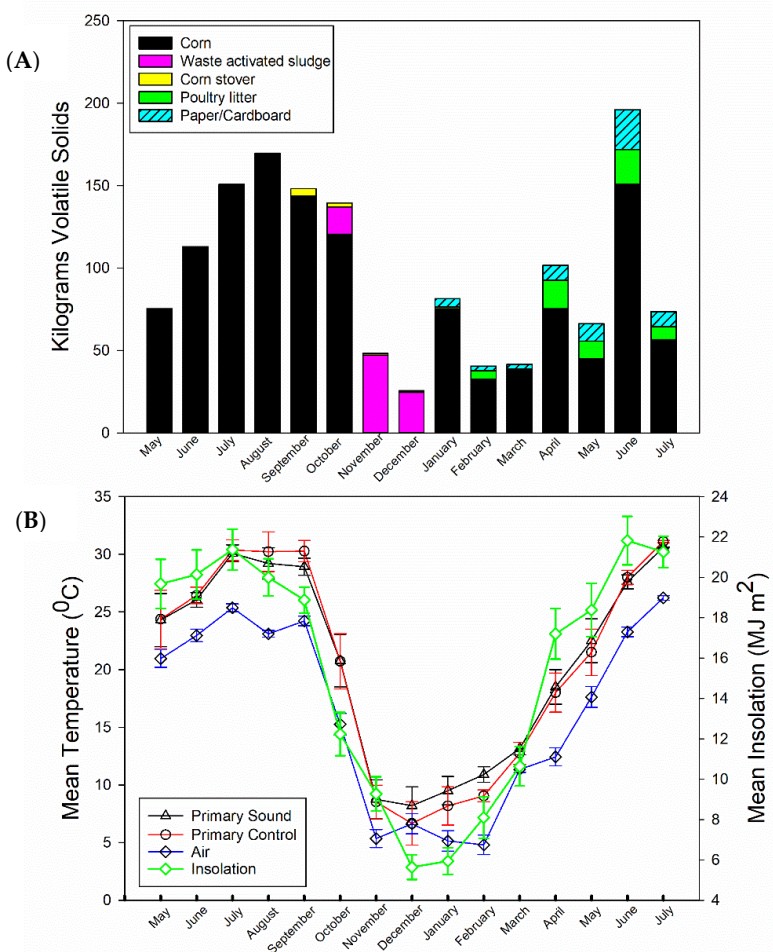

**Figure 2.** (**A**) Monthly feeding rate and source of volatile solids fed to each anaerobic digester system and (**B**) environmental conditions during the evaluation period.

## 2.2. Chemical Analyses

Wastewater and biogas analysis were carried out as described previously [15]. Organic carbon and nitrogen analysis were performed using a Shimadzu TOC-L instrument (Shimadzu Scientific Instruments, Columbia, MD) using Standards Methods 5310B for high-temperature catalytic oxidation [21]. Samples were diluted 10-fold to fit under the calibration curve using a Purelab® Flex 1 Ultra water purification system (ELGA, High Wycombe, United Kingdom) with a purity of 18.2 MΩ cm$^{-1}$.

Volatile fatty acids (VFA) were analyzed on a Varian 3800 gas chromatograph (GC) equipped with a flame ionization detector (FID) and a 30 m by 0.32 mm o.d. (outer diameter) Nukol™ (cross-linked, acid modified poly(ethylene glycol)) column with a 0.25 μm film thickness (Millipore Sigma, St. Louis, MO, USA). Wastewater samples were filtered through a 0.2 μm syringe filter and a 4.0-mL aliquot was acidified with 0.24 mL of 1.2 M HCl. Two μL of the samples were injected onto the GC operated as follows: injector temperature 200 °C, injector pressure 68.9 kPa, oven initial temperature 60 °C for 2 min, then programmed at 5 °C to 120 °C and then 10 °C to 200 °C. FID temperature was 220 °C, He makeup flow 25 mL min$^{-1}$, H$_2$ 30 mL min$^{-1}$, air 300 mL min$^{-1}$.

Inductively coupled plasma spectroscopy (ICP) was conducted after microwave-assisted acid digestion of sludge samples to determine metal and silicon concentrations using Standard Methods 3125 [21].

## 2.3. Statistical Analyses

Data for gas production, wastewater quality analyses, and dissolved gases were analyzed by one-way analysis of variance (ANOVA) using PROC ANOVA in the Statistical Analysis System

(SAS) system for Windows version 9.3 [22]. In comparisons of weekly gas production by primary sound-treatment versus control digesters, means were analyzed in SAS using PROC TTEST.

## 3. Results and Discussion

### 3.1. Digester Acoustics

As stated earlier, ultrasonification of waste is generally limited to pretreatment of waste to reduce sludge particle size and partially solubilize the waste, as high-frequency sound is attenuated in proportion to distance much more rapidly than is low frequency sound. In the gas-saturated environment of an anaerobic digester, and particularly within the sludge, however, a vast array of bubbles with a continuum of radii exist, all of which emit harmonics when excited at their resonant frequency [14]. Because of this, the possibility exists that ultrasonification of waste could be achieved in volumes otherwise impractical by conventional means by exploiting bubble harmonics.

In order to assess this potential, we excited the digester with a single frequency, 155.563 Hz (musical note D#3), which was chosen for its low frequency and for being above the lower limit of the speaker's claimed frequency range. The amplifier was operated at half-volume using the lower pair of speakers and recording was undertaken with the upper hydrophone. Spectral averaging was done on one minute of ambient sound in the digester, followed by one minute of excitation with the single frequency.

Figure 3 shows the amplitudes of the harmonics detected in the digester from $f_0$ (155.563 Hz) through $f_{137}$ (21,586 Hz). Higher harmonics could not be detected as this exceeded the recording capabilities of the audio equipment. Likely due to both a vast range of bubble sizes with matching resonant frequencies, as well as multiple harmonics emitted by the resonating bubbles, there was no drop off in the amplitude of these harmonics up to the range of low ultrasonic frequencies. This clearly shows that excitation of the gas saturated digestate with a low-frequency sound can cause ultrasonic emissions due to bubble harmonics at some distance from the sound source. As digesters typically develop distinct sludge and supernatant zones, this indicates that speakers could be placed within the supernatant and used to excite the greatest expanse of sludge while limiting sound attenuation due to suspended particles and bubbles [23]. Given the relatively small volume of the pilot-scale digesters, this was not considered as a problem, but in larger-scale digesters, speaker placement may be more critical.

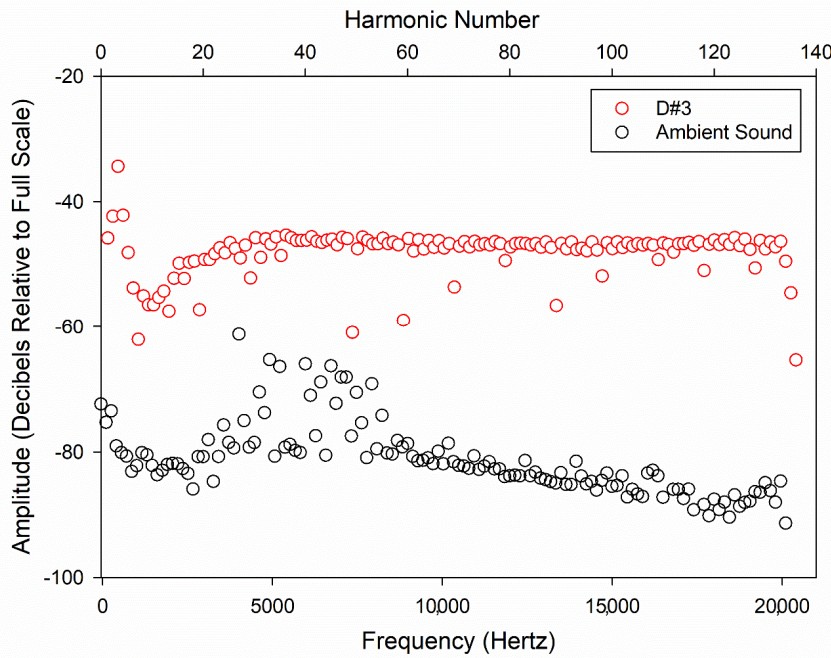

**Figure 3.** Peak amplitudes measured in primary sound-treated digester before and during exposure to single wavelength sound (D#3, 155.563 Hz). Peak amplitudes measured every 155 Hz.

As a continuous treatment, the digesters were exposed mostly to musical compositions. As explained in Loughrin et al. [15], we felt that variation in amplitude and frequency of the audio excitation would cause perturbations in the waste, leading to more cavitation inception and collapse. Although we could not ascertain the veracity of the assumption, it was easier to note cavitational events in recordings of digesters exposed to music. Also, in our previous study, we found that musical recordings gave greater frequency coverage than did sine waves even when accounting for bubble harmonics. MP3 recordings of the primary control digester and primary sound-treated digester are included in the Supplementary Materials.

The ratio of sound power levels between the control- and sound-treated digesters were compared using the bottom hydrophones and employing the formula:

$$Np = 10 \times \log\left(\frac{P}{P_0}\right) \tag{2}$$

where $P$ was the measured sound level in dB of the sound-treated digester and $P_0$ was the measured sound level in the control digester [24]. Using this formula, it was calculated that ambient noise levels in the sound-treatment were 22.6% higher than in the primary control, 68.4% higher than when playing simultaneous 150, 300, and 450 Hz sine waves with a 6 Hz tremolo to the primary sound-treatment, and 89.9% higher when playing Danse Macabre by Camille Saint-Saëns, both played at half volume. These ratios were calculated from recordings made on 14 August 2019. (Supplementary files: 1. Primary control bottom hydrophone amplified once at 400% ambient sount.mp3 and 2. Primary sound-treated bottom hydrophone amplified once at 400% ambient sound followed by the excerpt the planets (Holtz).mp3).

During colder weather (8 January 2020), it was calculated that ambient noise levels in the sound-treatment were 63.2% higher than in the control and during a loud passage of The Rite of Spring by Igor Stravinsky, sound levels were calculated to be 71.7% higher in the primary sound-treatment than in the primary control, again when the sound file was played at half volume.

These observations show that sound levels in the sound-treatment were much higher than in the control regardless of whether the speakers were in use. This was true even when the control, after accumulating large amounts of undigested waste during the winter (as discussed below), started to produce large volumes of gas during the second summer of operation. As the goal of using sound to treat the digester was to assist in sludge breakdown and assist in microbial utilization of the sludge, it was expected that the ambient sound level in the treated digester should be greater than in the control due to more cavitation events and bubble release. Although the primary control did produce many noticeable cavitation events, the amplitude of these were much lower than in the primary sound-treatment.

During warm weather, a regular clicking noise developed in the primary sound-treatment that was detected primarily by the lower hydrophone. Once this clicking noise developed, it was always present until temperatures cooled during October 2019 and did not reappear until the Spring of 2020. Using a recording of 14 August 2019 as an example, the clicking seemed to be composed of three broad peaks centered at about 3700, 2500, and 500 Hz and with a periodicity of approximately 1400 ms (Supplementary Materials 2. Primary sound-treated bottom hydrophone amplified once at 400% ambient sound followed by excerpt the planets (Holtz).mp3). When slowed down to 5% of normal speed, without changing the recording pitch, these broad peaks were resolved into what appeared to be a harmonic series of reverberating peaks centered about 2000 Hz, and spaced around 80 Hz apart, with more prominent peaks at approximately 3700, 3500, and 500 Hz. During warm weather, the clicking occurred with great regularity although occasionally low frequency cavitation events seemed to accelerate both the rate and amplitude of the clicking. Two examples of the clicking occurring on 9 August 2019 (at approximately 5.25 and 9.06 min) are also given as a supplementary recording (Supplementary Materials 3. 9 August 2019 Primary sound trt bottom hydrophone amp 1X 400% clicking rate affected by cavitation.mp3) as well as a recording of the clicking played at normal

speed and repeated at a speed of 5% (Supplementary Materials 4. 14 August 2020 Primary sound bottom hydrophone amp 2X 400% clicking normal speed repeated at 5% speed.mp3).

On 13 August, the clicking was eliminated upon feeding of the digester and had not returned when monitored about two h later. The following morning, the clicking was again present. Sound amplitude from 0–20 kHz as measured with the lower hydrophone were 9.7 dB lower after feeding then before. This was most likely due to bubble popping and loss of dissolved gases during feeding. This does not help to define what caused the clicking noise in the digester but does suggest that gas-saturated conditions were required.

Although the physics of cavitation dynamics and bubbles is not within the authors' area of expertise, we believe the clicking noise was caused by the release of gas from the sludge in a regular manner as bubble formation occurred at favorable nucleation sites. We feel the vibratory aspect of the signal was due to rapid bubble oscillations following bubble inception. Alternatively, it is possible this clicking is due to cavitational collapse, and the shock wave which then results. It is interesting to note that the residue remaining after the collapse of a bubble can act as a nucleation sites for formation of new bubbles [25].

### 3.2. Biogas Production and Wastewater Quality

Figure 4 shows the average daily biogas production for each month from the primary and secondary digester tanks for both systems. Over an evaluation period encompassing June 2019 through July 2020, biogas production averaged 16.6 $m^3$ and 13.1 $m^3$ week$^{-1}$ from the primary sound-treated and control digesters, respectively (Table 1), with much of the gas production from the primary control digester occurring in the last two months. The gas concentrations of $CO_2$ and $CH_4$ were also somewhat higher from the sound-treated digester than those of the control digester.

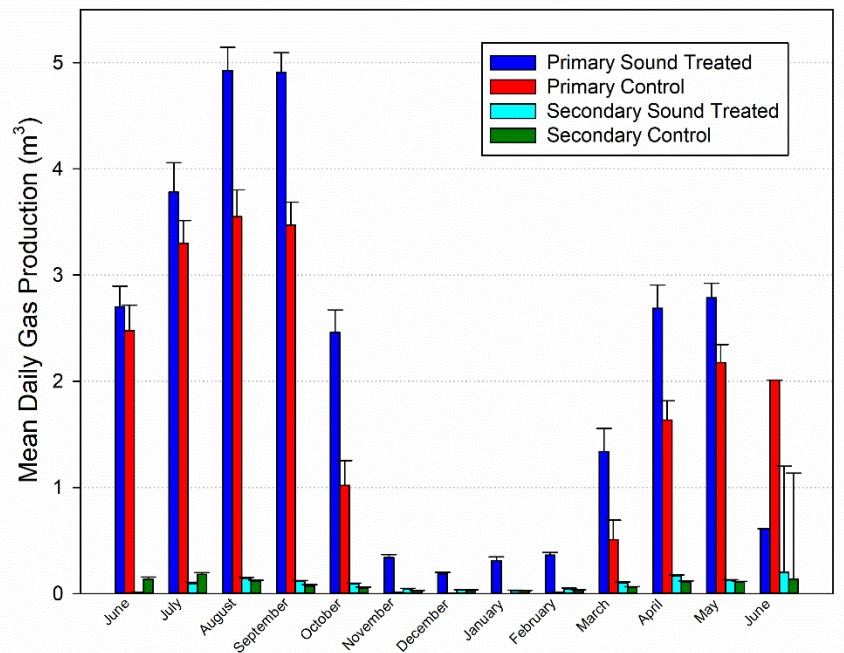

**Figure 4.** Daily gas production sound-treated and control digester systems. Data represent the mean of daily determinations for each month with standard error of the mean.

**Table 1.** Biogas and wastewater characteristics of control and sound-treated digester systems [a].

| Physiochemical Characteristic | Control | | | Sound-Treated | | |
|---|---|---|---|---|---|---|
| **Biogas** | **Primary** | **Secondary** | **Tertiary** | **Primary** | **Secondary** | **Tertiary** |
| Weekly gas production ($m^3$) | 13.1 ± 1.69 | 0.631 ± 0.063 | NA [b] | 16.6 ± 1.67 * | 0.653 ± 0.054 | NA |
| $CO_2$ (mmol $m^{-3}$) [c] | 10,300 ± 258 | 4540 ± 173 | NA | 10,900 ± 222* | 5760 ± 299 * | NA |
| $CH_4$ (mmol $m^{-3}$) [c] | 27,400 ± 716 | 8570 ± 904 | NA | 28,900 ± 665 * | 12,600 ± 1170 * | NA |
| **Wastewater** [d] | | | | | | |
| $HCO_3^-$ (mM) | 21.0 ± 0.35 | 21.4 ± 0.34 | 19.7 ± 0.33 | 20.9 ± 0.50 | 22.4 ± 0.31 * | 21.1 ± 0.24 * |
| $CO_3^{2-}$ (mM) | 0.018 ± 0.001 | 0.0360 ± 0.008 | 0.114 ± 0.015 | 0.019 ± 0.001 | 0.027 ± 0.002 | 0.074 ± 0.008 |
| $sCO_2$ (mM) | 4.25 ± 0.029 * | 2.39 ± 0.14 | 1.21 ± 0.14 | 3.54 ± 0.20 | 2.59 ± 0.13 * | 1.24 ± 0.12 |
| $sCH4$ (mM) | 23.0 ± 1.00 * | 24.3 ± 0.98 * | 18.7 ± 1.81 | 19.8 ± 0.98 | 22.0 ± 0.85 | 18.6 ± 1.22 |
| Temperature (°C) | 20.1 ± 1.23 | 19.9 ± 1.25 | 19.3 ± 1.19 | 20.4 ± 1.12 * | 19.5 ± 1.19 | 18.6 ± 1.19 |
| pH | 7.09 ± 0.03 | 7.34 ± 0.03 | 7.82 ± 0.07 | 7.15 ± 0.03 * | 7.32 ± 0.02 | 7.72 ± 0.05 |
| Turbidity (FNU) [e] | 591 ± 30.8 | 381 ± 37.7 | 265 ± 20.1 | 609 ± 30.2 | 398 ± 37.2 | 246 ± 15.4 |
| Specific conductivity (µS $cm^{-1}$) | 6860 ± 231 | 6880 ± 232 | 6210 ± 180 | 6980 ± 197 * | 7170 ± 176 * | 6610 ± 176 * |
| Chemical oxygen demand (mg $L^{-1}$) | 2650 ± 203 * | 1650 ± 112 | 1450 ± 103 | 2440 ± 178 | 1590 ± 94.6 | 1370 ± 88.7 |
| Suspended solids (mg $L^{-1}$) | 1660 ± 233 | 574 ± 49.9 | 479 ± 44.6 | 1610 ± 243 | 541 ± 45.1 | 463 ± 51.7 |
| Organic carbon (mg $L^{-1}$) | 1030 ± 59.3 * | 714 ± 23.6 | 597 ± 18.4 | 927 ± 44.5 | 685 ± 20.8 | 595 ± 16.3 |
| Total nitrogen (mg $L^{-1}$) | 1130 ± 16.8 | 1120 ± 14.5 | 1020 ± 20.4 | 1160 ± 15.4 * | 1170 ± 14.8 * | 1070 ± 17.8 * |

[a] Data represent the mean ± standard error of the mean of 59 once weekly determinations from June 2019 to mid-July 2020 when feeding of digesters ceased. In a paired comparison of specific stages of control and sound-treated systems, values labelled with an asterisk are significantly higher by analysis of variance of the $log_{10}$-tranformed values at $p = 0.05$. [b] NA = not applicable. [c] $n = 58$ once weekly determinations. [d] $n = 56$ once weekly determinations. [e] Formazin nephelometric unit.

Gas production from the secondary digesters was low, averaging only 3.9% and 5.2% that of the primary sound-treatment and control, respectively, while methane production was less than 2% that of the primary digesters. Gas production in the secondary digesters was likely low due to retention of solids within the primary digesters. Otherwise, it should be noted that environmental conditions such as pH, $HCO_3^-$ buffering, and temperature were also favorable for biogas production within the secondary digesters.

Biogas production from the primary sound-treatment digester was consistently higher than that of the primary control from June 2019 and continuing until the spring 2020. In late May 2020, as average daily temperature passed 20 °C, the primary control digester started to produce more biogas than did the primary sound-treatment.

During the same time period, gas production from the primary sound-treatment started to decline precipitously, and by early June was only about 20% that of the primary control. Still, up to the beginning of May 2020, cumulative gas production from the primary sound-treatment was 245,000 L more than that of the primary control. Since both control and sound-treated secondary digesters produced relatively little gas, it is safe to assume that the primary control had accumulated considerable undigested feed.

Given the decline in gas production from the primary sound-treatment and taking into consideration warm temperatures and stable pH of the primary digesters, feed amounts were increased in June 2020 (Figure 2). While this increased biogas production in the primary sound-treatment (Figure 4), gas production from the primary control continued to outpace it. Given the presumption of accumulated solids in the control digester and warming temperature, this result was expected.

Nominally dissolved gases ($sCH_4$, $sCO_2$) were high in all three digester tanks for both systems. These gases are referred to here as nominally dissolved since, at 20 °C and 10% salinity, $CH_4$ has a solubility of only about 1.5 mM [26]. This, together with $sCO_2$ concentrations, showed that the aqueous phase gases for the most part did not exist in a solvated state but rather as bubbles and what might be referred to as nascent, nano- or proto-bubbles (i.e., with a radius on the order of 1 nm [27]). Given that gas production occurs in the sludge layer and the majority of nucleation sites for gases can also be

assumed to occur there, sonic excitation of the sludge should result in numerous cavitation events which will help to break down and mix solids.

In order to determine if gas production from the digesters was dependent on sound intensity, the digesters were exposed to four or five week-long evaluation periods starting 9 July 2019 and concluding in early October with the amplifier volume at one-quarter, one-third, or one-half volume. These comparisons were confounded by the degree of sludge degradation within the digesters as well as variability in environmental conditions during each evaluation period. As regards the latter, daily air temperature averaged 25.1 ± 1.63, 24.3 ± 2.28, and 24.3 ± 4.77 °C, respectively, for the one-quarter, one-third, and one-half volume evaluation periods, and average daily insolation averaged 21.1 ± 2.24-, 20.4 ± 1.86-, and 17.4 ± 2.94-MJ m$^{-2}$ during these same periods.

When exposed to sound at one-quarter volume, the primary treated digester averaged 32.3 m$^3$ biogas per week, an increase of 27% over that of the primary control (Figure 5). When sound was played at one-third volume, the primary sound-treatment averaged 39% more gas production, and when played at one-half volume, the sound-treatment averaged 54% more gas production than did the control. From then on, the amplifiers were operated at one-half volume.

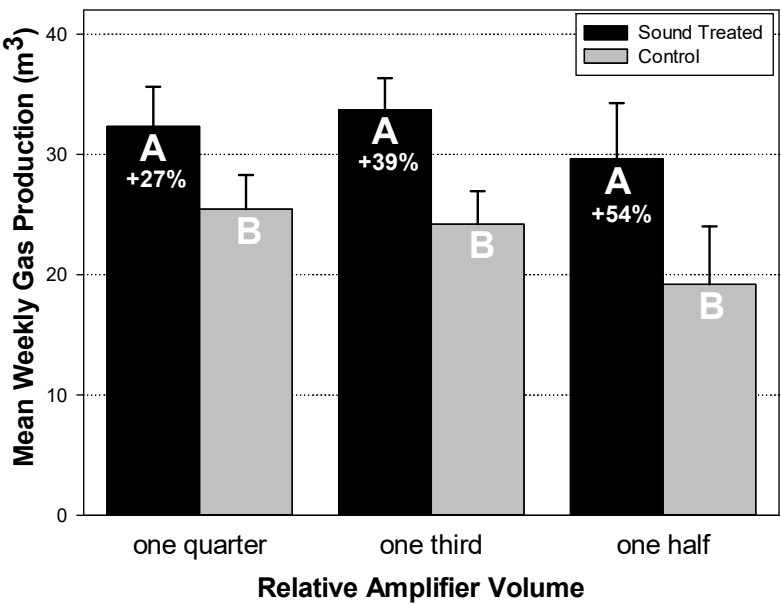

**Figure 5.** Comparison of daily gas production between primary sound-treatment and control at three volume levels. Data represent the mean ± standard error of the mean for four-week evaluation periods (one-quarter and one-half volume) and one five-week evaluation period (one-third volume). Within each evaluation period, bars labelled with different letters are significantly different by a Student's *t*-test at *p* = 0.05.

As discussed in the Introduction, there are a number of mechanisms that may be responsible for sound-enhancement of biogas production. The relative contribution of these individual mechanisms is unclear. It is likely, though, that all play a significant role in initial fragmentation of sludge, leading to higher microbial growth.

*3.3. Seasonal Performance*

Pham et al. [28] showed that temperatures above 20 °C are critical for maintaining gas production from unmixed anaerobic digesters. From 1 October to 8 October 2019, temperatures dropped from 29.8 °C to 20.5 °C in the primary control and from 29.3 °C to 21.5 °C in the primary sound-treatment and weekly gas production in the control and sound-treatment dropped by 77% and 49%, respectively. From that point onwards, as temperatures continued to slowly decline, gas production dropped and

some aspects of wastewater quality deteriorated. Table 2 shows gas production and wastewater quality during a period of warm weather (2 July 2019 to 1 October 2019) and cool weather (15 October to 4 March 2020) for the primary control and sound-treatment.

**Table 2.** Comparison of environmental conditions, gas production and wastewater quality between primary control and sound-treated digesters during warm and cool seasons.

| Environmental Parameter | Weather | | | |
|---|---|---|---|---|
| | Warm [a] | | Cool [b] | |
| Average daily air temperature (°C) | 21.1 ± 0.27 | | 5.8 ± 0.73 | |
| Average daily insolation (MJ m$^{-2}$) | 19.1 ± 0.48 | | 7.4 ± 0.52 | |
| Physiochemical characteristic | Control [c] | Sound-treated [c] | Control [d] | Sound-treated [d] |
| Wastewater temperature (°C) | 26.2 ± 0.43 | 26.0 ± 0.87 | 8.4 ± 0.76 | 9.6 ± 0.70 |
| Biogas | | | | |
| Weekly gas production (m$^3$) | 18.6 ± 1.56 | 25.4 ± 1.79 * | 0.027 ± 0.009 | 1.98 ± 0.195 * |
| $CO_2$ (mmol m$^3$) | 11,100 ± 143 | 11,500 ± 162 * | 8180 ± 596 | 9450 ± 563 |
| $CH_4$ (mmol m$^3$) | 30,400 ± 793 | 31,100 ± 795 | 22,300 ± 1450 | 26,000 ± 1670 |
| Wastewater | | | | |
| $HCO_3^-$ (mM) | 21.6 ± 0.34 | 21.6 ± 0.81 | 20.2 ± 0.68 | 20.5 ± 1.0 |
| $CO_3^{2-}$ (mM) | 0.029 ± 0.003 | 0.031 ± 0.002 | 0.019 ± 0.001 | 0.020 ± 0.006 |
| $sCO_2$ (mM) | 3.88 ± 0.46 | 3.08 ± 0.28 | 5.23 ± 0.60 | 4.22 ± 0.41 |
| $sCH_4$ (mM) | 21.3 ± 0.90 * | 18.8 ± 0.72 | 29.1 ± 1.83 * | 26.8 ± 2.16 |
| pH | 7.36 ± 0.01 | 7.41 ± 0.02 | 7.01 ± 0.06 | 7.06 ± 0.05 |
| Turbidity (FNU) [e] | 480 ± 36.7 | 458 ± 53.2 | 606 ± 63.4 | 550 ± 36.7 |
| Specific Conductivity (µS cm$^{-1}$) | 8350 ± 151 | 8260 ± 139 | 5000 ± 134 | 5420 ± 116 * |
| Chemical oxygen demand (mg L$^{-1}$) | 1660 ± 122 | 1390 ± 164 | 2660 ± 324 | 2650 ± 280 |
| Organic carbon (mg L$^{-1}$) | 816 ± 20.0 | 739 ± 21.2 | 1020 ± 99.9 | 980 ± 81.3 |
| Total nitrogen (mg L$^{-1}$) | 1180 ± 29.6 | 1200 ± 32.6 | 1130 ± 22.1 | 1140 ± 16.9 |

[a] Data represent the mean of 112 daily determinations ± standard error of the mean; [b] Data represent the mean of 140 daily determinations ± standard error of the mean; [c] Data represent the mean of 16 once weekly determinations ± standard error of the mean; [d] Data represent the mean of 20 once weekly determinations ± standard error of the mean. Within a seasonal comparison of control and sound-treated digesters, means followed by an asterisk are significantly greater by an Analysis of Variance test at $p = 0.05$. [e] Formazin nephelometric unit.

Our results were similar to those of Pham et al. [28], and during the cool period gas production from the primary control was negligible, averaging only 0.1% that of warm weather. On the other hand, while gas production from the sound-treatment was low during the months of November through February, it averaged 74-fold greater that of the control. Table 3 shows the average gas production for the primary and secondary anaerobic digesters for four temperature ranges.

**Table 3.** Daily biogas production for each digester based on temperature range of average daily temperature.

| Air Temperature Range (°C) | Number of Days Observed | Control | | Sound-Treated | |
|---|---|---|---|---|---|
| | | Primary | Secondary | Primary | Secondary |
| | | Daily Gas Production (L) | | | |
| −10 to 0 | 14 | 3.9 | 11.1 | 244 | 22.5 |
| 0 to 10 | 118 | 169 | 38.2 | 635 | 51.7 |
| 10 to 20 | 99 | 972 | 74.5 | 1930 | 102 |
| 20 to 30 | 164 | 3610 | 138 | 3920 | 119 |

While the primary sound-treatment outperformed the primary control at all four temperature ranges, the greatest proportionate gains in biogas production were seen at temperatures below 20 °C.

Gas production was similar in the secondary sound-treatment and secondary control and did not show as pronounced a temperature dependence as did the primary digesters. This was probably due to a combination of sludge retention in the primary digesters limiting feed to the secondary digesters and the smaller mass of the secondary digesters allowing for faster solar heating on cool but sunny days.

The relatively high biogas production of the secondary digesters during cool weather may have also been due, at least in part, to pH. For instance, from 2 January to 11 March 2020, the pH of the primary control averaged 6.79 ± 0.05 and the pH of the secondary control averaged 7.05 ± 0.04. The optimal pH for methanogenesis is generally accepted to be from 6.7 to 7.4 [29], so that conditions in the secondary control were likely more optimal for methanogenesis.

To repeat, some aspects of water quality deteriorated during cool weather (Table 2). The pH of the primary digesters declined, which caused $HCO_3^-$ to decline and $sCO_2$ concentrations to increase. COD and organic carbon both increased. Concentrations of identified VFA, although quite variable during the cool months, increased by approximately 5-fold in the primary control and 7-fold in the primary sound-treatment (Table 4). Some of the variability in VFA concentrations might be attributed to reduced feeding during the winter months (Figure 2). When pH of the digesters declined, the amount of feed given to the digesters was reduced until the pH rose again, i.e., until VFA concentrations declined. This led to a cycle of varying pH with equally variable VFA concentrations. For example, in the primary sound-treatment, acetate concentrations averaged 780 mM at pH 6.8 and below and less than 1 mM above pH 7.0.

**Table 4.** Volatile fatty acid concentrations in primary digesters during periods of warm and cool weather [a].

| Volatile Fatty Acid (mM) | Primary Control Digester | | Primary Sound-Treated Digester | |
|---|---|---|---|---|
| | Warm Weather | Cool Weather | Warm Weather | Cool Weather |
| Acetic | 0.90 ± 0.23 | 4.59 ± 3.80 | 0.15 ± 0.23 | 3.47 ± 2.52 |
| Propanoic | 0.07 ± 0.07 | 0.31± 0.49 | 0.05 ± 0.08 | 0.58 ± 0.62 |
| *iso*-Butyric | 0.26 ± 0.42 | 0.35 ± 0.65 | nd [b] | 0.20 ± 0.43 |
| Butyric | nd | 1.11 ± 1.06 | nd | 0.60 ± 0.59 |
| Total identified | 1.23 | 6.36 | 0.65 | 4.85 |

[a] Data represent the mean ± standard error of the mean of 12 determinations for each season. [b] nd = not detected.

Hydrolysis is often considered the rate-limiting step in sewage treatment and anaerobic digestion [13,28]. Hence, if polymeric substances such as polysaccharides and proteins within the sludge are not hydrolyzed to simple sugars and amino acids, fermentation and methanogenesis cannot occur. This is to some extent axiomatic, but results obtained here indicate accumulation of soluble substances, especially VFA, that seem to indicate reduced methanogenesis. Furthermore, it is easy to envision that if pH declines sufficiently due to VFA accumulation during cool weather digestion, this could negatively affect the growth of microorganisms and the activity of hydrolytic enzymes.

It should be incidentally noted that an effort was made to switch to feeding the digesters WAS from a municipal wastewater treatment system in late October (Figure 2). However, cool weather in January hardened the WAS, making feeding difficult, and with social distancing restrictions imposed due to COVID-19, WAS feeding was not resumed.

Other wastewater quality parameters that showed a marked difference between warm and cool seasons were dissolved gases ($CO_2$ and $CH_4$) and specific conductivity. As would be expected, dissolved gases were higher in concentration during the cool season due to increased solubility of gases at lower temperature. It is interesting to note that $sCO_2$ and $sCH_4$ were lower in the sound treatment regardless of season, likely indicating degassing of supernatant due to increased kinetic energy of the digestate. Specific conductivity, not surprisingly, decreased considerably during the cool season due to lower salt solubility in cooler water. In the fourth tanks which were aerated and used as to recirculate water back to the digesters for feeding, salt buildup occurred. This salt was analyzed by inductively coupled plasma spectroscopy (ICP) and was assayed as containing 790 g kg$^{-1}$ phosphate, 181 g kg$^{-1}$

Mg, and 8.5 g kg$^{-1}$ Ca with lesser amounts of Mn, Fe and Al. This agrees well with an identity as a struvite-like mineral. The pH of these tanks averaged 8.35 ± 0.083 and 8.39 ± 0.082 for the control and sound-treated systems, respectively, while carbonate ($CO_3{}^{2-}$) alkalinity averaged 0.25 ± 0.037 and 0.28 ± 0.038 µmol L$^{-1}$ (data not shown). Simoes et al. found that bacterial precipitation of struvite was optimal at a pH range of 7.3 to 8.3 [30]. Struvite accumulation in anaerobic digesters and wastewater treatment systems forms scales which can block pipes and reduce the efficiency of pumps. If, as was the case here, recirculation of wastewater through the digestion system is used, the means of collecting the struvite and other insoluble minerals should be used to maintain operational efficiency and reduce phosphorus loads.

### 3.4. Sludge Characteristics

Table 5 shows sludge characteristics averaged over the course of the experiment. In general, sludge VS content was lower in the primary sound-treatment than in the primary control. The only means available for sludge sampling was a drain port near the bottom of each tank. Thus, it was difficult to obtain sludge samples that we felt were representative of the overall digester contents and no significant differences in volatile solids, ash content, metal or metalloid (silicon) content were noted in comparisons of control and sound-treated digesters. Still, since average VS content in the primary sound-treatment was lower than that of the primary control and this likely indicates that for the majority of the experiment, sound treatment did accelerate sludge reduction compared to the control.

**Table 5.** Sludge characteristics of primary and tertiary digesters.

| | Primary Control | Tertiary Control | Primary Sound | Tertiary Sound |
|---|---|---|---|---|
| | Concentration in g kg$^{-1}$ digestate wet weight [a] | | | |
| Volatile solids | 75.8 ± 15.4 | 8.6 ± 3.1 | 54.1 ± 8.9 | 12.6 ± 3.3 |
| Ash | 28.1 ± 6.0 | 8.1 ± 2.7 | 19.4 ± 3.0 | 9.4 ± 2.7 |
| | Concentration in mg kg$^{-1}$ digestate wet weight [b] | | | |
| Calcium | 801 ± 206 | 218 ± 95.5 | 793 ± 127 | 218 ± 95.5 |
| Phosphorus | 799 ± 210 | 356 ± 170 | 704 ± 130 | 299 ± 88.7 |
| Aluminum | 759 ± 189 | 206 ± 88.8 | 739 ± 167 | 115 ± 33.2 |
| Potassium | 430 ± 32.6 | 349 ± 20.4 | 439 ± 35.6 | 377 ± 20.0 |
| Iron | 174 ± 50.3 | 53.3 ± 25.7 | 185 ± 45.9 | 47.2 ± 17.5 |
| Sodium | 140 ± 8.7 | 120 ± 8.2 | 143 ± 8.7 | 161 ± 16.2 |
| Magnesium | 138 ± 58.7 | 162 ± 80.1 | 96.7 ± 13.6 | 129 ± 38.8 |
| Silicon | 88.4 ± 26.6 | 31.7 ± 10.0 | 83.9 ± 16.8 | 38.6 ± 8.6 |
| Manganese | 34.0 ± 9.2 | 10.9 ± 5.6 | 31.7 ± 7.5 | 6.8 ± 2.3 |
| Zinc | 21.0 ± 4.1 | 6.2 ± 2.6 | 24.6 ± 3.7 | 10.1 ± 2.9 |
| Copper | 5.7 ± 1.2 | 1.7 ± 0.8 | 6.2 ± 0.9 | 2.7 ± 1.0 |
| Lead | 0.3 ± 0.1 | 0.1 ± 0.02 | 0.3 ± 0.1 | 0.1 ± 0.02 |
| Total | 3390 | 1515 | 3250 | 1404 |

[a] Data represent the mean of 11 once monthly determinations ± standard error of the mean. [b] Data represent the mean of 15 once monthly determinations ± standard error of the mean.

Unsurprisingly, the most abundant metals in sludge of the primary digesters were multivalent species such as Ca and P where, despite the mixing received by the tanks during feeding, concentrations were much higher than in the tertiary tanks likely due to precipitation. Concentrations of the monovalent ions K and Na were more equally distributed between the primary and tertiary stages of the treatment systems. Only a few ICP analyses were conducted on the secondary digesters, but results indicated that 32.4%, 42.6%, and 25% of quantified Ca precipitated out in the primary, secondary, and tertiary tanks, respectively, whereas for P, these values were 43.7%, 37.4%, and 19.0%. Wastewater pH is known to have a profound effect on metal solubility [31] and the tendency for successive stages of the treatment systems to increase in $CO_3{}^{2-}$ alkalinity (Table 1) probably accounts for the observed differences in concentrations of metals. Perhaps by manipulating the pH of anaerobic digesters, while remaining

within optimal range for methanogenesis, considerable amounts of metals of environment concern could be removed from wastewater.

The digesters were fed for the last time on 9 July 2020. Within one week, gas production by the primary sound-treatment had declined by 87% and by 63% from the primary control. The primary digesters were opened on 7 August and sludge samples were taken eight locations within the primary digesters.

Sludge from the primary sound-treated digester averaged $18.0 \pm 1.1$ g VS kg$^{-1}$, whereas sludge from the primary control averaged $15.3 \pm 2.8$ g VS kg$^{-1}$. Sludge ash averaged $28.9 \pm 2.4$ and $26.5 \pm 3.9$ g kg$^{-1}$ from the primary sound-treatment and control, respectively. At the beginning of May 2020, however, sludge VS from the primary sound treatment was measured at 46.3 g kg$^{-1}$, while sludge VS from the primary control was measured at 95.2 g kg$^{-1}$. Despite the difficulty in obtaining what we felt were representative sludge samples, this seems to account for the great increase in gas production by the primary control near the end of the experiment.

Given the rapid decline in gas production once feeding of the digester ceased and the low sludge VS content measured at the end of the experiment, it was obvious the digesters were operated well below optimal capacity during the last few months. Digester sludge was likely well colonized by microbes and since water temperature in the digesters were averaging 30 °C or more, considerably more feed could have been added to the digesters.

As Supplementary Materials, gas production during the final three months of the experiment along with ICP analysis of feedstock and sludge at the end of the experiment as determined at eight locations within the primary digesters is given (Feedstock and Final Sludge ICP.xlsx).

## 4. Conclusions

Sound can clearly be used to accelerate digester startup and substitute to an appreciable extent for heating requirements during cool weather. It also seems likely to allow higher waste loading rates although this was not tested in the present study. Further research may help to elucidate if sound treatment can be used at larger scale, if biogas production and sludge reduction can be optimized by the selection of frequency used, and if sound exposure should be continuous or if shorter exposure periods are adequate.

**Supplementary Materials:** The following are available online at http://www.mdpi.com/1996-1073/13/20/5349/s1: MP3 file 1: Primary control bottom hydrophone amplified once at 400% ambient sound; MP3 file 2: Primary sound-treated bottom hydrophone amplified once at 400% ambient sound followed by excerpt the planets (Holtz).mp3; MP3 file 3: 9 August 2019 Primary sound trt bottom hydrophone amp 1X 400% clicking rate affected by cavitation.mp3; MP3 file 4: 14 August 2020 Primary sound bottom hydrophone amp 2X 400% clicking normal speed repeated at 5% speed.mp3; Feedstock and Final Sludge ICP.xlsx.

**Author Contributions:** Conceptualization, J.L. and S.A.; methodology, J.L., N.L., S.A., and J.S.; validation, J.L., and S.A.; formal analysis, J.L.; investigation, J.L., S.A., J.S.; resources, J.L., K.S., and S.A.; data curation, S.A., and J.L., writing-original draft preparation, J.L.; writing-review and editing, J.L., S.A., K.S., N.L., J.S.; visualization, J.L., supervision, J.L., K.S.; project administration, J.L., and S.A.; funding acquisition, J.L., and K.S. All authors have read and agreed to the published version of the manuscript.

**Funding:** This research received no external funding and was conducted as part of USDA-ARS National Program 212: Soil and Water, Developing Safe, Efficient and Environmentally Sound Management Practices for the Use of Animal Manure.

**Acknowledgments:** The authors thank Mike Bryant and Zachary Berry (USDA-ARS) for technical assistance. Mention of a trademark or product anywhere in this article is to describe experimental procedures, does not constitute a guarantee or warranty of the product by the USDA, and does not imply its approval to the exclusion of other products or vendors that may also be suitable.

**Conflicts of Interest:** The authors declare no conflict of interest.

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
