# Peer review of "In Situ Sonification of Anaerobic Digestion: Extended Evaluation of Performance in a Temperate Climate"

_energies, doi:10.3390/en13205349_

Round 1

Reviewer 1 Report

The benefits of in situ sonification to anaerobic digestion process under seasonal temperature fluctuation are clearly presented, although the paper can be improved to better support the discussion.

Abstract: main quantitative results should be given in the abstract to support the conclusions.

Line 124: delete ‘the pressure of’.

Figure 2: (a) One symbol is missing from the legend of Volatile Solids figure: there are 6 symbols in the graph, but only 5 are explained in the legend; and (b) the application of in-situ sonification, e.g. sound intensity, over the course of experiment shall be given in Figure 2.

Line 194-196: (a) this should be section 2.3; and (b) there is no result or discussion on malodorous compounds in this paper.

Line 313-317: the explanation in this paragraph is confusing. Please explain further the logical relationship between favourable environmental conditions in secondary digesters and the retention of feedstock in primary digesters.

Table 1: (a) It is difficult to believe the data present in the table are the mean +/-standard error over one year. For instance, the temperature information given in the Table doesn’t reflect the temperature range between >30 and <10 shown in Figure 1; (b) Please explain why the dissolved CH4 is disproportionally higher than dissolved CO2, in comparison with their partial pressure in gas phase, i.e. headspace partial pressure, also in comparison with their solubility in water; (c) It is not clear why there is a subscript 4 for sCO2 and sCH4; and (d) the footnotes seem not to be complete. There is a subscript e without explanation.

Section 3.4: the sludge characteristics are discussed in this section, although sludge analysis is not explained in section 2.2.

Author Response

Report 1

The benefits of in situ sonification to anaerobic digestion process under seasonal temperature fluctuation are clearly presented, although the paper can be improved to better support the discussion.

Abstract: main quantitative results should be given in the abstract to support the conclusions.

We added a bit more detail to the abstract. We also changed the word exhaustion on line 25 to depletion since this seemed more descriptively accurate.

Line 124: delete ‘the pressure of’.

We deleted this. We’re sorry for the duplication.

Figure 2: (a) One symbol is missing from the legend of Volatile Solids figure: there are 6 symbols in the graph, but only 5 are explained in the legend; and (b) the application of in-situ sonification, e.g. sound intensity, over the course of experiment shall be given in Figure 2.

There are only 5 symbols in the graph. The paper/cardboard is one legend, perhaps the figure is confusing because the amounts of the various feedstocks vary so much it is difficult to view. We slightly modified the colors in the graph to be (hopefully) clearer. We also modified the figure to spell out terms like waste activated sludge because on reflection we felt that the table should be more self-explanatory.

  1. The sound intensity was one half volume throughout most of the experiment. We added a better explanation of the investigation into sound volume starting on line 378 and in the Material and Methods line 181. As we feel volume levels on an analog amplifier are not precise, and with our equipment not really quantifiable, we feel this is best left out of the graph. (the power output from the amplifier is affected by the size of the speakers (i.e. largely the mass of the magnet) so that power consumed is not a linear relationship with volume. We could modify the figure to include volume if the editors prefer.

Line 194-196: (a) this should be section 2.3; and (b) there is no result or discussion on malodorous compounds in this paper. I was working on a paper on malodors of animal waste (cresol, skatole, etc.) and must have been distracted. I apologize for this error going undetected and have corrected the text including the section numbering.

Line 313-317: the explanation in this paragraph is confusing. Please explain further the logical relationship between favourable environmental conditions in secondary digesters and the retention of feedstock in primary digesters.

We edited this paragraph so hopefully it is less confusing. The point we are trying to make is that we feel that solids must have been largely retained/consumed with the primary digesters, otherwise gas production would have been higher in the secondary digesters since environmental conditions were also favorable in the secondary digesters.

Table 1: (a) It is difficult to believe the data present in the table are the mean +/-standard error over one year. For instance, the temperature information given in the Table doesn’t reflect the temperature range between >30 and <10 shown in Figure 1; (b) Please explain why the dissolved CH4 is disproportionally higher than dissolved CO2, in comparison with their partial pressure in gas phase, i.e. headspace partial pressure, also in comparison with their solubility in water; (c) It is not clear why there is a subscript 4 for sCO2 and sCH4; and (d) the footnotes seem not to be complete. There is a subscript e without explanation.

  1. The data is correct. The standard errors seem a bit low, but this is due to a high number of observations. The tanks were also dark colored which undoubtably kept the water temperatures higher than they would have been during the winter months.
  2. Dissolved gas concentrations were calculated using Henry’s constants and the Henderson-Hasselbalch equation. While the situation for CH4 is fairly straight forward, CO2 is just one part of the bicarbonate buffering system:

Solvated CO2 ↔ H2CO3 (carbonic acid) ↔ HCO3- + H+

Using dimensionless Henry’s constants of 27 and 0.8 for CH4 and CO2 respectively, we calculated these values by injecting water samples into an acid solution to calculate total CO2 in the samples and then apportioned the form of the CO2, solvated, bicarbonate, and carbonate. The method is explained in Frontiers in Environmental Science 5:65 (2017).

  1. I apologize for this. I prepared the tables using numbered footnotes originally and then noticed that lettered footnotes are preferred for the journal. I neglected to delete the superscript “fours” and also was negligent in cleaning up the table as I was exploring different ways of inserting footnotes to make the table as neat as possible.
  2. The footnote e was just another mistake on my part made while editing the table.    

Section 3.4: the sludge characteristics are discussed in this section, although sludge analysis is not explained in section 2.2. We added a sentence to section 2.2.

Reviewer 2 Report

Manuscript by Loughrin et al. presents some unique insights into the impacts of sound on the performance of anaerobic digestion conducted through the seasonal variations. The paper may be of interest for Energy's readers; however, it still contains unclear statistical expressions and several typo-errors. I would recommend acceptance of this work after a thoroughly major revision. Some of my comments can be found below. 

General comments:

1) The use of past participles as adjectives throughout the manuscript is not harmonised. For example, sound-treated digester (has a hyphen), but most in the abstract and introduction, no hyphen is used. Please check throughout your paper.

2) The term 'digester' and 'digesters' have been used for 160 times in your paper. However, many of them can be omitted or shortened, e.g., 'the primary control digester' can be simple as 'the primary control,' the primary sound-treated digester' can be simple as 'the primary sound treatment.'  

Line 21: Through the late spring 2019 through April 2020?

Lines 30-31: Correct the grammar.

Line 56: As noted by Yu et al. [7], ...

Figure 1: It's good to have this figure; however, it's hard to see in details. Would it be possible to enlarge or zoom up what you are emphasising?

Line 139: Use the right degree for Celsius.

Figure 2: Use the right degree for Celsius in the y-axis. Indicate the title of the x-axis. In the figure legend, describe the period (Month Year - Month Year) of the experiments/measurements. Denote the upper graph with A and the lower one with B, and describe the plots are means derived from how many replications; also, the error bars represent what? Also, explain all about primary sound, primary control, air, and insolation; what does each of these parameters mean? 

Lines 175-176: A close parenthesis is missing.

Line 178: use 'o' letter, not '0' number zero for 'of.'

Line 186: Use multiply symbol, not x letter and superscript TM of Nukol.

Line 187: .25-μm

Line 188: 0.2-μm ... 4.0-mL

Lines 190-191: Use the right degree for Celsius.

Line 191: Remove dot behind min.

Line 195: Provide the full term of PROC ANOVA. If it is a commercial name of the software, please define which statistics you used, e.g., one-way analysis of variance (ANOVA), etc.

Line 210: What is (D#3)?

Line 216: 137 should be subscript?

Line 199: It seems that you mixed Results with Discussions. Therefore, change the title of this section.

Line 232: Again, what is (D#3)?

Line 247: Subscript 0.

Line 280: Remove dot behind min.

Line 281: What is trt?

Line 283: What do you mean by 4.?

Line 296: Correct the grammatical errors.

Line 305: ... concentrations ...

Line 306: ... than those from ... Check your English usage.

Figure 4: Why the three initials are used here for indicating the months, which differs from those used in Figure 2? Please unite your writing. The means derived from how many replications? Why standard errors, not standard deviation? Why didn't you compare these means using your statistics?

Line 315: What do you mean by 'this'? Unclear, please clarify. 

Line 320: Use the right degree for Celsius.

Line 343: Remove.

Table 1: You can use the abbreviation for every unit presented in this Table, e.g., Millimoles = mmole, Micromolar = μM. Why standard errors, not standard deviations? The statistical comparisons look strange. I wonder how did you perform a pairwise comparison with ANOVA, as ANOVA requires at least triplicate values for the comparison. What kind of ANOVA you used? Did you check the normal distribution of your data? Use the right degree for Celsius. What is NA in the Table? What is for 'e' notation? Please move Biogas Characteristics, Gas Concentration (Millimoles m-3), Wastewater Concentration (Micromolar), Wastewater Characteristics, and Concentration (Milligrams per Liter), to the left-hand side of the Table. Do those n = 58 or 56 come from a single sampling and you measured 58 or 56 times, or you made multiple samplings with those n(s), please clarify? What are sCO2 4 and sCH4 4, please explain in the Table's footnote? Formazin nephelometric unit, in singular? A close parenthesis is missing for Specific Conductivity.

Line 364: Use the right degree for Celsius. Sometimes 'percent,' sometimes '%,' which one you prefer? 

Line 365: What do you mean by 'this'?

Lines 375-378: average daily air temperature averaged? Please revise.

Figure 5: 'One' for all at the x-axis. Why standard error, not standard deviation? What is the error bar, a half SE? Student's t-test is not included in 2.2. What about the statistical comparison across different relative amplifier volumes?

Line 394: Why didn't you check the microbial growth? It is not that difficult.

Lines 398, 400: Use the right degree for Celsius.

Line 399: 8 October, which year?

Line 402: Remove the hyphen behind 77.

Table 2: Please improve your Table in the same way as mentioned for Table 1.

Table 3: Move 'Daily gas production (L)' to the top row of the Table. Use the right degree for Celsius. How many measurements you did?

Line 427: Use the right degree for Celsius.

Line 436: those conditions?

Table 4: Remove Concentration (Millimolar), and add (mM) behind Volatile Fatty Acid. Why standard error? Why didn't you provide the statistical comparisons?

Lines 456-458: Please check your English usage.

Line 472: What do you mean by 'this'?

Line 474: A space in front of alkalinity.

Table 5: Put 'Sludge characteristics' at the top left row and move the units to the left (use acronyms for the units). What standard errors? Why didn't you provide the statistical comparisons?

Line 500: 31.%?

Line 510: Which year?

Line 522: How did you know that it has been colonised well by microbes? Use the right degree for Celsius.

Author Response

Manuscript by Loughrin et al. presents some unique insights into the impacts of sound on the performance of anaerobic digestion conducted through the seasonal variations. The paper may be of interest for Energy's readers; however, it still contains unclear statistical expressions and several typo-errors. I would recommend acceptance of this work after a thoroughly major revision. Some of my comments can be found below. 

 General comments:

  • The use of past participles as adjectives throughout the manuscript is not harmonised. For example, sound-treated digester (has a hyphen), but most in the abstract and introduction, no hyphen is used. Please check throughout your paper.

We made this more consistent.

 2) The term 'digester' and 'digesters' have been used for 160 times in your paper. However, many of them can be omitted or shortened, e.g., 'the primary control digester' can be simple as 'the primary control,' the primary sound-treated digester' can be simple as 'the primary sound treatment.'  

 We changed this throughout the manuscript.

Line 21: Through the late spring 2019 through April 2020?

 As per our response to reviewer one’s comments, we changed this sentence.

Lines 30-31: Correct the grammar.

The grammar has been checked.

 Line 56: As noted by Yu et al. [7], ...

We modified the placement of the reference number.

 Figure 1: It's good to have this figure; however, it's hard to see in details. Would it be possible to enlarge or zoom up what you are emphasising?

Line 139: Use the right degree for Celsius. We changed the symbol used throughout the manuscript.

 Figure 2: Use the right degree for Celsius in the y-axis. Indicate the title of the x-axis. In the figure legend, describe the period (Month Year - Month Year) of the experiments/measurements. Denote the upper graph with A and the lower one with B, and describe the plots are means derived from how many replications; also, the error bars represent what? Also, explain all about primary sound, primary control, air, and insolation; what does each of these parameters mean? 

Lines 175-176: A close parenthesis is missing.

 We put the close parenthesis in.

Line 178: use 'o' letter, not '0' number zero for 'of.'

 We corrected this.

Line 186: Use multiply symbol, not x letter and superscript TM of Nukol.  We applied the superscript to TM of Nukol (and did this also for ® symbol and replaced the letter x with the word “by” which might be less confusing that either the x or times symbol.

Line 187: .25-μm We corrected this.

 Line 188: 0.2-μm ... 4.0-mL We also corrected this.

 Lines 190-191: Use the right degree for Celsius. As per the reviewer’s previous comments, we replaced all the superscript 0’s with the degree symbols.

 Line 191: Remove dot behind min. We removed it. 

Line 195: Provide the full term of PROC ANOVA. If it is a commercial name of the software, please define which statistics you used, e.g., one-way analysis of variance (ANOVA), etc. We expanded this sentence.

 Line 210: What is (D#3)? D#3 is the musical note (155.563 Hertz) we used to explore harmonics in the digester (Figure 3). We added the words “musical note” before the first occurance of D#3.

 Line 216: 137 should be subscript? The reviewer is correct so we fixed that.

 Line 199: It seems that you mixed Results with Discussions. Therefore, change the title of this section. We changed it. 

Line 232: Again, what is (D#3)? We addressed this by adding the words “musical note” 3 comments earlier.

 Line 247: Subscript 0. We replaced this.

 Line 280: Remove dot behind min. We deleted the period.

 Line 281: What is trt? Trt is used as an abbreviation in the file name just to keep it a little bit shorter as amp 2X is used as an abbreviation for amplified two times.

Line 283: What do you mean by 4.? This is just part of the file name.

 Line 296: Correct the grammatical errors. We modified this paragraph in an attempt to make the meaning clearer.

Line 305: ... concentrations We changed this to plural. ...

 Line 306: ... than those from ... Check your English usage. We modified the sentence.

 Figure 4: Why the three initials are used here for indicating the months, which differs from those used in Figure 2? Please unite your writing. The means derived from how many replications? Why standard errors, not standard deviation? Why didn't you compare these means using your statistics?

Line 315: What do you mean by 'this'? Unclear, please clarify. This paragraph was modified per the query of reviewer #1.

 Line 320: Use the right degree for Celsius. We changed the symbol throughout the manuscript.

 Line 343: Remove. We’re not sure what is referred to here. Is it the blank part of the page resulting from Table 1 not being able to fit on the space left on page 9?

Table 1: You can use the abbreviation for every unit presented in this Table, e.g., Millimoles = mmole, Micromolar = μM. Why standard errors, not standard deviations? The statistical comparisons look strange. I wonder how did you perform a pairwise comparison with ANOVA, as ANOVA requires at least triplicate values for the comparison. What kind of ANOVA you used? Did you check the normal distribution of your data? Use the right degree for Celsius. What is NA in the Table? What is for 'e' notation? Please move Biogas Characteristics, Gas Concentration (Millimoles m-3), Wastewater Concentration (Micromolar), Wastewater Characteristics, and Concentration (Milligrams per Liter), to the left-hand side of the Table. Do those n = 58 or 56 come from a single sampling and you measured 58 or 56 times, or you made multiple samplings with those n(s), please clarify? What are sCO2 4 and sCH4 4, please explain in the Table's footnote? Formazin nephelometric unit, in singular? A close parenthesis is missing for Specific Conductivity.

We prefer to spell out the units in the table in that it doesn’t save any space in the paper as does using abbreviations in the text body. We just feel that for the headers, there’s no strong reason to prefer abbreviations in place of spelling the words out. We used anova since the data was balanced, and his all the comparisons were simply comparing the two means, an ANOVA will basically give the same results as a t-test. The data did meet the criteria for normality. We changed the symbol for degrees throughout the manuscript. The data came from weekly determinations over the course of the experiment. We added the word weekly three times to the footnotes to make this clearer. We suppose that referring to Formazin Nephelometric units in the plural is correct but ask the Editor’s opinion on the matter. We tried moving the headers such as “wastewater characteristics” to the left side of the table but it seemed to us to give a “cluttered” look and made the table harder to read. As all the cells in those rows are merged, it would be easy for the editors to move the headings to the left hand side if they prefer.

 Line 364: Use the right degree for Celsius. Sometimes 'percent,' sometimes '%,' which one you prefer? We changed all the occurances of percent to %.

 Line 365: What do you mean by 'this'? we’re not sure what is being referred to here but we did correct the grammar at this point by replacing the word “as” with “are” (now line373).

 Lines 375-378: average daily air temperature averaged? Please revise. We corrected this error.

 Figure 5: 'One' for all at the x-axis. Why standard error, not standard deviation? What is the error bar, a half SE? Student's t-test is not included in 2.2. What about the statistical comparison across different relative amplifier volumes? We added the student’s ttest to the material and methods section. We did not compare gas production across the different volumes used as the season was progressing and cooling temperatures had a large influence on gas production. This can be noticed in Figure 5 especially for the control digester.

Line 394: Why didn't you check the microbial growth? It is not that difficult. We have been attempting to get our Unit’s microbiologist to do that for us but as we are a small Unit, it hasn’t been done so far. We have collected and preserved samples under his directions, so hopefully it will be done in the future.

 Lines 398, 400: Use the right degree for Celsius. We changed this throughout the manuscript.

 Line 399: 8 October, which year? We modified the sentence to be clearer (now line 207).

 Line 402: Remove the hyphen behind 77. Did this (line 409).

 Table 2: Please improve your Table in the same way as mentioned for Table 1. We clarified the number of determinations in the footnotes.

 Table 3: Move 'Daily gas production (L)' to the top row of the Table. Use the right degree for Celsius. How many measurements you did? We modified the table.

 Line 427: Use the right degree for Celsius. . We changed this throughout the manuscript.

Line 436: those conditions? The phrase now occurs on line 442, we’re more comfortable with the word “that”.

 Table 4: Remove Concentration (Millimolar), and add (mM) behind Volatile Fatty Acid. Why standard error? Why didn't you provide the statistical comparisons? We can change the word millimolar to mM if the editor prefers. We prefer to keep it unabbreviated as, in being column headers, it doesn’t take up more space.

 Lines 456-458: Please check your English usage. We checked the grammar here and it seems correct. We did, however, delete the word “would”. 

Line 472: What do you mean by 'this'? We added the word “salt” after “this”.

 Line 474: A space in front of alkalinity. We did this change.

 Table 5: Put 'Sludge characteristics' at the top left row and move the units to the left (use acronyms for the units). What standard errors? Why didn't you provide the statistical comparisons? As per our previous answers, we prefer to keep the headers centered unless the editor prefers them left justified.

We used standard errors throughout the manuscript instead of standard deviations since a standard deviation is an indication of the variability of the data around the mean while the standard error is an indication of how far the mean is likely to be from the true population mean. In a related manner, regarding the graphs, especially for Figure 2, using standard deviations would have made panel B very difficult to read. The only practical alternative for that graph would have likely entailed leaving standard deviations or errors out of the graph to keep it legible.

We found no statistically significant differences for the data in Table 5 and indicated this starting on line 493.    

Line 500: 31.%? We fixed this error.

Line 510: Which year? The year was indicated, 2020.

Line 522: How did you know that it has been colonised well by microbes? Use the right degree for Celsius. This sentence was modified since our statement that the sludge was well colonized by microbes is supposition. As per our previous responses, we changed the symbol for degree throughout the manuscript.

Round 2

Reviewer 2 Report

I don't see any changes in the submitted manuscript file but see some in the highlighted version requested. I commented about the units used in the Tables and the Table presentation. The authors defended to use as they are presented because there is space around to use the units' full terms. I don't see this is a reasonable answer, and I don't think it's too hard for you to improve the way to present your data. To make your paper more consistent, international, and organized, please use SI base unit symbols. All Tables need significant revision before publication. My comments and suggestion can be found in the attached file.

Author Response

We have changed the tables as per the reviewer’s request. We apologize if the previous versions were not clear enough. The units for specific conductivity are correct (microSiemens per centimeter). The gas production units are expressed as cubic meters per week in tables 1 and 2 and liters per day in table 3 under the convention trying to keep numbers from being either too large or small. If we had expressed gas production as cubic meters per day during the cool season it would have resulted in very small numbers (e.g. 0.0039 m3 per day instead of 3.9 L per day).

The units for specific conductivity are correct. We have made table 1 and 2 agree on the term “specific conductivity”. We did check the turbidity and have added a row to table 2. We made an error labelling bicarbonate, carbonate, etc. concentrations as micromolar in table 1 and have changed the units. We apologize. We have labeled  nitrogen as total nitrogen in table 1 and 2. We changed the column reading tank temperature to wastewater temperature.

On table 3, we changed the format of the table to the reviewer’s suggestion.

We also changed the format of table 4 to the reviewer’s suggestion except that we did not put in the number of days observed as there were 12 observations for both cool and warm seasons as noted in the footnote.

We changed table 5 so that both ash and elemental composition read g (mg) kg-1 digestate wet weight.

Round 3

Reviewer 2 Report

The manuscript is well improved but still contains some typo errors. The authors should proofread thoroughly and correct the errors throughout the paper before publication.

Author Response

We thank the reviewer for their trouble through multiple rounds of edits. We have made the following changes:

We resolved all previous changes and deleted previous yellow highlighting to make the changes done during this round of edits clear.

List of changes made during this round:

Line 68,  136, 156, 288, 297 -deleted extra space between sentences or after commas.

Line 289 deleted space before a comma.

Line 69 – changed “disruption of” to “disrupting” to make sentence grammatically correct.

Line 242 – we changed “quantify the veracity” to “ascertain the veracity” since this seemed a better choice of words.

Line 280, we inserted the word “approximately” to make it agree with the statement on line 275 about the frequency of the audio peaks.

On line 341, we inserted the words “control and sound treated” before the word digesters since otherwise the sentence seemed a bit vague.

On line 354, we inserted a closing parenthesis.

On line 387, we inserted the % symbol after 77.

From line 377 onwards, we changed the line spacing from 17 point to single to make it consistent with previous paragraphs.

Regarding Table 2, we hope you the editor's fix the placement of the table. We corrected the placement of the numbers in the row for Chemical oxygen demand by placing hard returns on the numbers.

We also changed the line spacing and paragraph indents throughout the manuscript to make them consistent. This we did not highlight and apologize for the inconsistencies in formatting.